# ⛛ Fuse2Match: Training-Free Fusion of Flow, Diffusion, and Contrastive Models for Zero-Shot Semantic Matching

**Jing Zuo[1]    Jiaqi Wang[1]    Yonggang Qi[1] ✉    Yi-Zhe Song[2]**
[1]School of Artificial Intelligence, Beijing University of Posts and Telecommunications
[2]SketchX, CVSSP, University of Surrey
zuoj0723@gmail.com, {wang_jiaqi,qiyg}@bupt.edu.cn, y.song@surrey.ac.uk
✉ Corresponding author

## Abstract

Recent work shows that features from Stable Diffusion (SD) and contrastively pretrained models like DINO can be directly used for zero-shot semantic correspondence via naive feature concatenation. In this paper, we explore the stronger potential of Stable Diffusion 3 (SD3), a rectified flow-based model with a multimodal transformer backbone (MM-DiT). We show that semantic signals in SD3 are scattered across multiple timesteps and transformer layers, and propose a multi-level fusion scheme to extract discriminative features. Moreover, we identify that naive fusion across models suffers from inconsistent distributions, thus leading to suboptimal performance. To address this, we propose a simple yet effective confidence-aware feature fusion strategy that re-weights each model's contribution based on prediction confidence scores derived from their matching uncertainties. Notably, this fusion approach is not only training-free but also enables per-pixel adaptive integration of heterogeneous features. The resulting representation, Fuse2Match, significantly outperforms strong baselines on SPair-71k, PF-Pascal, and PSC6K, validating the benefit of combining SD3, SD, and DINO through our proposed confidence-aware feature fusion. Code is available at https://github.com/panda7777777/fuse2match

## 1  Introduction

Dense correspondence prediction is a fundamental computer vision task that aims to establish pixel-level matches between two images. Conventional methods often rely on a large amount of task-specific data and fine-tuning. In contrast, humans can naturally align semantically related regions without supervision. This has motivated growing attention in zero-shot semantic dense correspondence, which serves as a critical benchmark for assessing the generalizability of large-scale foundation models.

Recent advancements in large-scale diffusion models for text-to-image generation have sparked interest in their potential beyond generative tasks. In particular, works such as DIFT [27] and SD+DINO [33] demonstrate that features extracted from diffusion models (i.e., Stable Diffusion [25]), either alone or in combination with contrastive models (i.e., DINO [3]), enable strong performance in zero-shot semantic correspondence, without any fine-tuning. However, existing research typically adopts UNet-based Stable Diffusion and has not fully explored the latest architectures or the question of how to effectively combine multiple pretrained models. Specifically, we ask: (i) *Can the latest flow-based diffusion models, such as SD3, bring further benefits?* (ii) More importantly, *can we*

39th Conference on Neural Information Processing Systems (NeurIPS 2025).

*better integrate diverse pretrained models with complementary strengths for dense matching, in a training-free way?*

A common strategy for model fusion is knowledge distillation [11], where a student network learns to mimic multiple teacher models, such as AM-RADIO [24]. While effective, distillation requires costly training, and often fails to preserve the diversity of the source models. In this work, we introduce a confidence-aware feature fusion framework that is training-free: features from different off-the-shelf models are fused at inference time, based on their estimated matching confidence. This allows us to leverage complementary inductive biases across different model families (e.g., contrastive [3] vs. generative pertaining [25], ViT [5] vs. UNet [26] vs. DiT [21] backbones), in a lightweight and effective manner.

Importantly, our work is not a simple extension of prior methods like SD+DINO or DIFT. While we draw inspiration from these studies, we tackle a much more challenging and unexplored scenario: the use of rectified flow models with transformer-based backbones (i.e., MM-DiT in SD3) for zero-shot correspondence. Due to substantial differences in architecture and training objectives (e.g., flow-matching vs. denoising), extracting meaningful features from SD3 is highly non-trivial. Following previous wisdom to naively select a single timestep and transformer block from SD3 even leads to inferior performance compared to UNet-based SD, despite SD3 being a more powerful model overall. Through a systematic exploration of hyperparameters such as timesteps, transformer blocks, and attention facets (e.g., key, query, value, token), we reveal that semantic information is dispersed across multiple blocks and facets in SD3. Thus, we explored multi-level fusion across timesteps and layers, enabling SD3 to achieve performance on par with DIFT (SD) and DINOv2.

In addition, we investigate whether incorporating features from earlier UNet-based Stable Diffusion (SD) models and contrastively pretrained ViTs (i.e., DINO) can further enhance the SD3 representations. However, our empirical analysis reveals substantial inconsistencies across these foundation models, stemming from their heterogeneous architectures and training objectives. As a result, naive feature concatenation leads to suboptimal performance due to conflicting feature distributions. To address this, we propose a simple yet effective confidence-aware fusion strategy that adaptively re-weights each model's contribution in a per-pixel fashion, guided by a matching uncertainty-based confidence score. This design differs from SD+DINO, which performs feature fusion by treating each model's contribution equally across all pixels. Moreover, we show that our fusion mechanism is generalizable across different model combinations, demonstrating its flexibility and robustness.

Our main contributions are as follows: (i) To our best knowledge, it is the first attempt to leverage SD3, a large-scale text-to-image generation pretrained model built on rectified flow, for zero-shot semantic correspondence without any task-specific fine-tuning. (ii) A simple yet effective method is devised for pixel-wise adaptive fusion of features from diverse foundation models, leveraging their complementary knowledge arising from distinct architectures and pretext training tasks. As a result, robust and generalizable features for dense matching across categories, viewpoints, and domains are obtained. (iii) Substantial performance gains have been achieved across various benchmarks (SPair-71k, PF-Pascal, and PSC6K datasets), validating the effectiveness.

## 2    Related Work

**Zero-Shot Semantic Correspondence.**    The goal of semantic correspondence is to identify matching object locations that share the same semantics, regardless of differences in categories[27], viewpoints[6, 18], deformations [8], or domains[16]. Many recent works [2, 27, 33] focus on the zero-shot setting, which requires no task-specific fine-tuning. This is achieved by directly utilizing the features obtained from foundation models, which are shown to be effective in learning implicit semantic correspondence. DINOv1 was first explored by Amir et al. [2] as a feature extractor with localized semantic information, demonstrating its applicability to various zero-shot tasks such as segmentation and correspondence. DINOv2[20], trained on larger and higher-quality datasets, has been shown to achieve superior performance in zero-shot correspondence tasks [33]. Text-to-image diffusion models, such as DIFT, have been leveraged for semantic correspondence lately since stable diffusion (SD) features have a strong sense of spatial layout, thereby facilitating part-level correspondence. SD+DINO demonstrates that SD and DINO features have different properties that are complementary. By simply concatenating these two features, significant performance improvements can be achieved. In this work, we show that the latest flow-based model, i.e., SD3, trained by text-to-image generation

can tackle zero-shot semantic correspondence. In addition, we observe significant performance gains over SD+DINO by fusing features of SD3, SD, and DINO.

**Representation Learning with Diffusion Models.** Diffusion models have demonstrated exceptional performance in image and video generation [7]. By generative modeling trained using large-scale text-photo pairs, diffusion models have been shown to be an effective representation learner exhibiting robust generalization to novel scenarios [30]. It has been observed that the intermediate layers of the UNet decoder can capture rich semantic information across various granularity levels by utilizing different timesteps and layer depths during denoising [27, 29]. Consequently, representations derived from diffusion models have proven to be highly effective in downstream tasks involving image segmentation [28], classification [31], and semantic correspondence [27, 33]. In this work, we focus on directly utilizing features obtained from multimodal diffusion transformers (MM-DiT) for zero-shot semantic correspondence, which is less studied.

## 3 Preliminary on Stable Diffusion 3

We begin by introducing background on Stable Diffusion 3 (SD3), which serves as the feature extractor for semantic correspondence prediction.

**Rectified Flow Model.** One of the key properties of Stable Diffusion 3 (SD3) [7] is that it is built upon Rectified Flow (RF) [1, 14, 15], which connects data $p_0$ and noise distribution $p_1 = \mathcal{N}(0, I)$ via linear paths, hence can mitigate error accumulation during sampling. Coupled with the novel Logit-Normal sampler, SD3 outperforms early versions of diffusion models, such as SDXL [22]. The forward process is defined as:

$$z_t = (1 - t)x_0 + t\epsilon, \tag{1}$$

where $x_0$ is the data sample, $\epsilon \sim \mathcal{N}(0, I)$ is Gaussian noise, and $t \in [0, 1]$ denotes the timestep. The backward process reverses this trajectory using a velocity field $v_\Theta(z_t, t)$, governed by the ordinary differential equation (ODE):

$$\frac{dz_t}{dt} = v_\Theta(z_t, t). \tag{2}$$

The model is trained via the Conditional Flow Matching (CFM) [12], which minimizes the discrepancy between the learned velocity field $v_\Theta(z_t, t)$ and the target velocity field $u_t(z_t|\epsilon)$:

$$\mathcal{L}_{\text{CFM}} = \mathbb{E}_{t, z_t, \epsilon} \|v_\Theta(z_t, t) - u_t(z_t|\epsilon)\|_2^2. \tag{3}$$

**MM-DiT Backbone.** Unlike the commonly adopted UNet architecture in the early versions of Stable Diffusion, another key difference in SD3 is the employed diffusion transformer backbone DiT. Notably, DiT backbone exhibits better scalability and facilitates bidirectional information exchange between text and image modalities, resulting in fine-grained, semantically rich image representations.

## 4 Methodology

In this section, we first introduce how we extract pixel-level matching features from Stable Diffusion 3 and utilize them to perform the nearest search for semantic correspondence. We then introduce our method to fuse features from diverse large-scale pretrained models.

### 4.1 Problem Setup

Following the common practice [2, 27, 33], semantic correspondence aims to find paired locations that share similar semantics between two images. We simply extract dense pixel features in both photos then match them. Given a feature map $F_i$ for image $I_i$, the feature vector for pixel at location $p$ is $F_i(p)$. Then we can match the most relevant pixel from image $I_2$ given a pixel $p_1$ from image $I_1$, which can be formally defined as:

$$p_2 = \arg \min_p d(F_1(p_1), F_2(p)) \tag{4}$$

where $d$ denotes the cosine distance used to mature the feature vector similarity.

## 4.2 Extracting Matching Features From MM-DiT

**Forward Process.** Similar to the Latent Diffusion Models (LDM) [25], given an image $x_0 \in \mathbb{R}^{h \times w \times 3}$, we first project it into the latent space through a pretrained VAE, i.e., $z_0 = \mathcal{E}(x_0)$, where $z_0 \in \mathbb{R}^{H \times W \times D}$ represents the latent image, and $\mathcal{E}$ denotes the pretrained encoder. Gaussian noise is then added to corrupt $z_0$ at a noise level determined by timestep $t$ using Eq. (1). Next, MM-DiT blocks are used to predict noise at each timestep.

**Feature Extraction via Backward Process.** Given the corrupted latent image $z_t$, we can extract features from the MM-DiT blocks of the pretrained SD3 during the backward denoising process. Apart from $z_t$ and timestep $t$, a text prompt embedding is fed into the network as well for image denoising. Following DIFT, we simply adopt "A photo of a {object category}." as the text prompt, which is encoded into an embedding $c \in \mathbb{R}^{L_c \times D_c}$ using pretrained, frozen text encoders. To construct the image embedding, we first add positional embedding and patchify the latent image $z_t$, then project this patch encoding and the text embedding to the same dimension. The text and image embeddings are concatenated as a sequence, which is then processed by the MM-DiT blocks to predict the injected noise at timestep $t$. Notably, each modality is handled by its own independent transformer, yet the input sequence contains both text and image

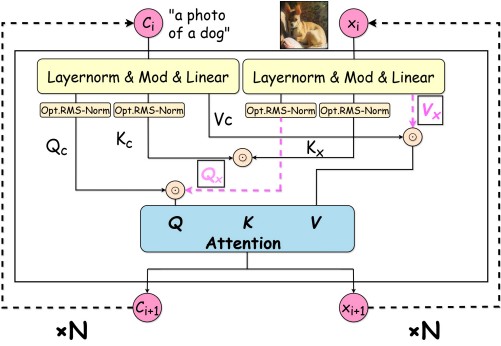

Figure 1: We extract SD3 features from intermediate MM-DiT blocks and empirically identify that the query ($Q_x$) and value ($V_x$) from the image-branch attention blocks serve as the most effective features for semantic matching.

tokens. This allows each representation to operate within its own domain while still exchanging knowledge from the other modality.

**Feature Facets Selection.** While also built upon the ViT architecture, SD3 provides a broader set of feature facets by allowing cross-modal interactions. As shown in Figure 1, it feeds both a text prompt embedding (i.e., $c_i$) and an image embedding (i.e., $x_i$) into the transformer blocks, enabling the emergence of self-attention, cross-attention, and joint attention, in addition to the conventional feature facets used in DINO-ViT [2], such as tokens, queries, keys, and values. In this context, self-attention refers to interactions among tokens within the same modality (e.g., image-to-image or text-to-text), while cross-attention facilitates information exchange between different modalities (e.g., text-to-image). Joint attention denotes the process of concatenating text and image embeddings prior to attention computation, enabling unified reasoning over fused representations. To identify the most effective representations for semantic matching, we perform a greedy search to select the optimal feature facets for establishing correspondences between two images. Note that once the hyperparameters are determined via greedy search on the training set, they remain fixed during inference, introducing no additional computational overhead.

## 4.3 Confidence-aware Feature Fusion

We hypothesize that different large-scale pretrained models, including DINOv2, UNet-based Stable Diffusion (SD), and DiT-based Rectified Flow Transformers (SD3), all learn implicit and complementary knowledge of semantic correspondence due to their diverse network architectures and pretraining tasks. SD and SD3 perform latent space generative pertaining, while DINOv2 is obtained via contrastive learning. As recent evidence shows that simply fusing features from SD and DINO yields surprisingly strong performance in zero-shot semantic correspondence. Therefore, we are interested in exploring whether combining SD3 with other pretrained models would provide additional benefits for zero-shot semantic correspondence.

A straightforward approach is to directly concatenate SD3 features with those from other pretrained models, such as DINO and SD. However, due to fundamental differences in network architectures and training objectives, foundation models naturally exhibit diverse feature characteristics when performing semantic matching, which inevitably leads to inconsistencies across models. To address

this, we propose a confidence-aware feature fusion strategy that effectively integrates the most reliable predictions from each model. Intuitively, we place greater trust in models that exhibit lower uncertainty in their semantic correspondences between two images. In practice, given a pair of image feature maps, i.e., $\boldsymbol{F_1} \in \mathbb{R}^{h_1 \times w_1 \times C}$ and $\boldsymbol{F_2} \in \mathbb{R}^{h_2 \times w_2 \times C}$ extracted from a pretrained model, a feature vector of query pixel $p_1$ at location $(x, y)$ in $F_1$ is defined as $\boldsymbol{v}_{x,y} \in \mathbb{R}^C$. Then a similarity vector $M \in \mathbb{R}^{h_2 w_2}$ representing the distances between $p_1$ and all locations in $F_2$ can be obtained by:

$$M = \boldsymbol{v}_{x,y} \cdot \texttt{flatten}(\boldsymbol{F_2})^T \tag{5}$$

where $\texttt{flatten}(\boldsymbol{F_2})$ is a flattened matrix derived from feature map $\boldsymbol{F_2}$, containing $h_2 w_2$ pixel vectors. Consequently, we define the prediction certainty by measuring the gap between the maximum and average value in the similarity vector $M$:

$$w = \max_{x,y} \boldsymbol{M} - \text{avg} \, \boldsymbol{M} \tag{6}$$

Intuitively, a sharp activation in $\boldsymbol{M}$ suggests high confidence in matching, whereas a more evenly distributed similarity across $\boldsymbol{M}$ implies lower confidence. To this end, we fuse features from different pretrained models by concatenating their re-weighted features with the normalized confidence scores. This ensures that models with higher matching confidence contribute more prominently, with the fusion operating at the pixel level to allow for highly localized and adaptive feature integration.

## 5 Experiments

### 5.1 Experimental Settings

**Datasets.** We evaluate the MM-DiT features and its fused features on three challenging benchmarks, namely SPair-71K [19], PF-Pascal [10], and PSC6K [16]. Following the tuning-free setting, we directly test our pixel matcher on the corresponding test sets. SPair-71K and PF-Pascal are datasets constructed for photo-to-photo matching. SPair-71K contains 12,714 test image pairs across 18 categories, featuring a wide variety of object keypoints and viewpoints, which pose significant challenges for dense matching. PF-Pascal, on the other hand, has 299 test image pairs distributed across 20 categories. PSC6K is a photo-sketch paired dataset consisting of 6,250 photo-sketch pairs across 125 categories, with 150,000 keypoint annotation pairs. Establishing semantic correspondence between photos and sketches is particularly challenging since free-hand sketches are inherently abstract, lacking the texture and color cues present in RGB images, which are crucial for accurate dense correspondence prediction. Evaluation on the PSC6K dataset is significant as it verifies whether pretrained models trained with photorealistic images can generalize to vastly different domains, such as human-drawn sketches.

**Competitors.** We compare our method against the current state-of-the-art semantic correspondence approaches, specifically focusing on those in the zero-shot setting, where point matching is performed directly based on their distance in the feature maps of image pairs, without additional task-specific fine-tuning, involving SD+DINO [33], DIFT [27], OpenCLIP[23], DINOv2[20], DINOv1[3], and SigLip[32]. To gain insights about their comparison results over conventional unsupervised and supervised methods with task-specific fine-tuning, we also include ISCVGSM[18], ASIC[9], CATs++[4], DHF[17] and SD4Match[13] for comparison.

**Evaluation Metric.** Following [27, 33], we adopt the Percentage of Correct Keypoints (PCK) metric to assess the accuracy of keypoint correspondences. Essentially, a predicted keypoint is deemed correct if it is sufficiently close to the corresponding ground truth keypoint, i.e., the distance is within $\tau \times \max(h, w)$, where $\tau$ is a positive scalar in the range [0, 1] controlling the strictness of the criterion, and $(h, w)$ denote the dimensions of the object bounding boxes in the SPair-71k dataset, or the image resolutions in the PF-Pascal and PSC6K dataset.

**Implementation details.** We implement our approach using PyTorch. We choose the Stable Diffusion 3.5 large model for extracting MM-DiT features. During feature fusion, the Stable Diffusion 2.1 and DINO v2 large models are adopted following the experimental setups in [27, 33]. In accordance with the common practice, we retain the original image resolutions used by each foundation model during training, i.e., 768 x 768 for SD, 480 x 480 for DINOv1, 840 x 840 for

Table 1: Comparison results (PCK@0.1) on SPair-71k dataset. **S**, **U**, **ZS**$^S$, and **ZS**$^M$ represent supervised, unsupervised, zero-shot using single model, and zero-shot with multiple models, respectively. Our proposed SD3-based approaches are color-coded in gray. The best and second-best scores are marked in **bold** and underlined, respectively.

| | Method | Aero | Bike | Bird | Boat | Bottle | Bus | Car | Cat | Chair | Cow | Dog | Horse | Motor | Person | Plant | Sheep | Train | TV | All |
|---|---|---|---|---|---|---|---|---|---|---|---|---|---|---|---|---|---|---|---|---|
| **S** | CATs++ | 60.6 | 46.9 | 82.5 | 41.6 | 56.8 | 64.9 | 50.4 | 72.8 | 29.2 | 75.8 | 65.4 | 62.5 | 50.9 | 56.1 | 54.8 | 48.2 | 80.9 | **74.9** | 59.9 |
| | DHF | 74.0 | 61.0 | 87.2 | 40.7 | 47.8 | 70.0 | **74.4** | 80.9 | 38.5 | 76.1 | 60.9 | 66.8 | 66.6 | **70.3** | 58.0 | 54.3 | **87.4** | 60.3 | 64.9 |
| | SD4Match | 75.3 | 67.4 | 85.7 | 64.7 | 62.9 | 86.6 | 76.5 | 82.6 | 64.8 | 86.7 | 73.0 | 78.9 | 70.9 | 78.3 | 66.8 | 64.8 | **91.5** | 86.6 | 75.5 |
| **U** | ISCVGSM | 74.8 | 64.5 | 87.1 | 45.6 | 52.7 | **77.8** | 71.4 | **82.4** | 47.7 | 82.0 | 67.3 | **73.9** | 67.6 | 60.0 | 49.9 | **69.8** | 78.5 | 59.1 | 67.3 |
| | ASIC | 57.9 | 25.2 | 68.1 | 24.7 | 35.4 | 28.4 | 30.9 | 54.8 | 21.6 | 45.0 | 47.2 | 39.9 | 26.2 | 48.8 | 14.5 | 24.5 | 49.0 | 24.6 | 36.9 |
| **ZS**$^S$ | SigLip | 51.5 | 41.7 | 74.4 | 25.1 | 34.1 | 41.1 | 42.7 | 61.1 | 23.8 | 49.1 | 47.5 | 46.1 | 41.4 | 58.4 | 25.7 | 41.3 | 45.8 | 18.7 | 42.8 |
| | DINOv1 | 41.3 | 27.5 | 67.2 | 23.1 | 45.3 | 39.7 | 35.8 | 65.5 | 25.8 | 55.1 | 45.9 | 31.8 | 27.5 | 44.5 | 34.1 | 38.7 | 45.7 | 42.0 | 42.5 |
| | DINOv2 | **75.5** | **67.5** | 86.4 | **47.9** | 46.2 | 58.0 | 53.9 | 71.0 | 37.8 | 69.5 | **67.8** | 70.0 | 69.6 | 67.0 | 30.9 | 65.6 | 56.9 | 31.6 | 59.0 |
| | OpenCLIP | 53.2 | 33.4 | 69.4 | 28.0 | 33.3 | 41.0 | 41.8 | 55.8 | 23.3 | 47.0 | 43.9 | 44.1 | 43.5 | 55.1 | 23.6 | 31.7 | 47.8 | 21.8 | 41.4 |
| | DIFT (SD2) | 63.2 | 54.3 | 80.3 | 34.4 | 46.0 | 52.2 | 48.3 | 77.1 | 38.9 | 76.6 | 55.2 | 61.4 | 53.2 | 45.8 | 57.6 | 56.3 | 70.9 | 63.5 | 59.4 |
| | SD3 | 63.7 | 50.9 | 79.4 | 37.0 | 51.9 | 58.1 | 52.8 | 73.9 | 41.9 | 70.7 | 52.2 | 55.1 | 57.2 | 60.4 | 53.0 | 54.6 | 63.8 | 64.6 | 59.8 |
| **ZS**$^M$ | SD+DINO | 73.4 | 64.0 | 86.5 | 39.8 | 53.0 | 55.2 | 53.9 | 78.4 | 46.3 | 77.6 | 65.0 | 69.8 | 63.2 | 69.1 | 58.9 | 68.0 | 66.7 | 54.5 | 63.3 |
| | SD3+DINOv2 | 75.5 | 66.4 | 86.8 | 45.1 | **58.5** | 61.5 | 56.2 | 77.0 | 48.5 | 76.6 | 64.9 | 69.2 | **69.7** | 69.0 | 57.2 | 66.3 | 64.5 | 62.9 | 65.9 |
| | SD+SD3+DINOv2 | 74.3 | 63.8 | 86.5 | 43.0 | 56.1 | 61.6 | 56.8 | 80.2 | **52.1** | 81.3 | 65.3 | 70.0 | 67.4 | 68.1 | **66.2** | 67.4 | 74.2 | 69.8 | 67.9 |
| | Fuse2Match | 75.1 | 66.2 | **87.7** | 43.4 | 55.3 | 61.4 | 57.1 | 81.1 | 51.9 | **82.1** | 67.1 | 71.6 | 67.9 | 68.8 | **66.2** | 69.6 | 75.4 | 69.9 | **68.8** |

DINOv2, and 1024 x 1024 for SD3. Given an input image in resolution $h \times w$, the extracted features using those models will be upsampled by bilinear interpolation into an image size feature map, resulting in feature map of size $(h \times w \times 9728)$ for SD3, $(h \times w \times 1280)$ for SD, and $(h \times w \times 1024)$ for DINOv2. All those features are normalized before concatenation, resulting in the combined feature map of size $(h \times w \times 12{,}032)$, substantially larger than those used in DIFT and SD+DINO. To reduce the computational cost, we apply matrix factorization, which significantly improves efficiency and allows inference on a single A100 40G GPU.

## 5.2 Semantic Correspondence Results

**Quantitative Results on SPair-71k.** Following the same setting in DIFT and SD+DINO (i.e., SD-2.1+DINOv2 more precisely), the metric PCK@0.1 (i.e., setting $\tau = 0.1$) is reported on the SPair-71k dataset. Results are shown in Table 1, we can see that: (i) SD3 (Ours) can achieve slightly better results (PCK@0.1 59.8) among competitors using features from single pretrained models, where OpenCLIP, SigLip, and DINOv1/v2 demonstrate relatively weaker performance compared to DIFT and SD3, suggesting the advantages of generative diffusion and flow models over discriminatively or contrastively trained models. (ii) Feature fusion using UNet-based diffusion and DINOv2 (i.e., SD+DINO) achieves PCK@0.1 score of 63.3, clearly outperforming all single pretrained models. Compared to SD+DINO, feature fusion using SD3 and DINO yields consistently improved performance (63.3 vs. 65.9), highlighting the superior compatibility and representational capacity of SD3 in conjunction with DINO. Interestingly, this advantage emerges despite SD3 and SD-2.1 achieving comparable results when used individually, suggesting that the improvements stem not from raw performance differences, but from enhanced complementarity in the fused representation. (iii) The highest performance is obtained by integrating SD3, stable diffusion (SD), and DINO via our proposed confidence-aware fusion strategy. We attribute this gain to the architectural heterogeneity between SD3 (DiT-based) and earlier SD variants (UNet-based), which enables more complementary feature representations for semantic matching. Notably, the fusion scheme guided by estimated confidence scores (i.e., Fuse2Match) consistently outperforms naive feature concatenation (i.e., SD + SD3 + DINOv2), underscoring the effectiveness of our adaptive weighting mechanism.

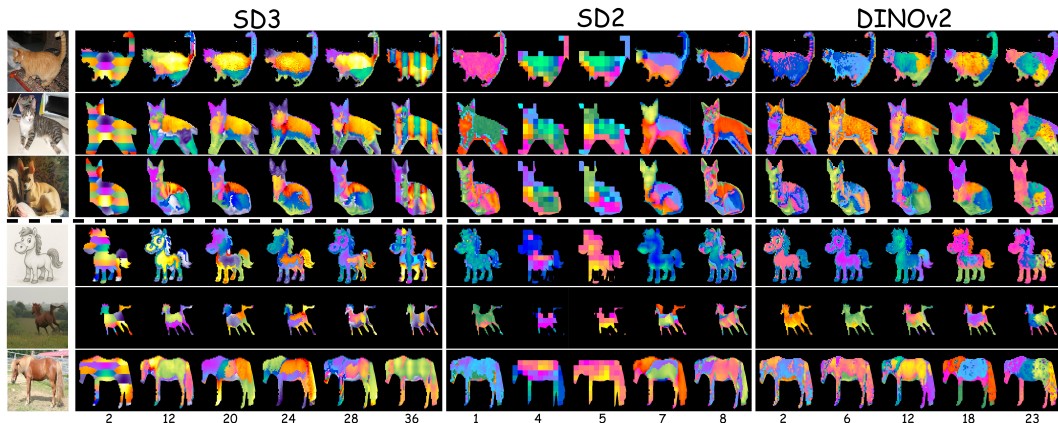

Figure 2: Visualization of feature maps from SD3, SD, and DINO via PCA and K-Means clustering.

*'Conflict' and 'complementary' effects among foundation models.* We evaluate (i) whether different single foundation models can achieve correct matching at the same locations, and (ii) whether the combination of foundation models can achieve improved per-point matching results compared to single models, to explicitly study their conflict and complementary effects. The idea is that the matched locations would be different when conflicts among foundation models exist, and improved matching results are expected due to the complementary effects offered by feature combination, either via naive concatenation or our proposed confidence-aware feature fusion. Specifically, given the test set of SPair-71k, which contains 12,234 image pairs, we evaluate whether each foundation model (i.e., SD3, SD, DINOv2) can achieve correct matching for at least half of the annotations within each image pair. As shown in Table 2, we present the number of image pairs that could be correctly matched by different numbers of foundation models. For example, none of the models succeeded in matching 1,984 image pairs (denoted as None), while only one model succeeded in another 1,974 image pairs, and so forth. It is evident that models often conflict with each other, as some models fail at specific matchings while others succeed, indicating contradictory predictions. Regarding the complementary effects among foundation models, we can see that, for example, simply concatenating features from all models improves correct matches from none to 6.42%, which further increases to 12.75% when adopting our confidence-aware feature fusion approach, validating their complementary effects. Similar trends can be observed even though saturation occurs in easier cases, such as the 5,719 image pairs where all single foundation models succeed. In addition, we visualize the feature maps from SD3, SD, and DINO using PCA and K-Means clustering. The resulting clusters shown in Figure 2 demonstrate that each model captures the data with distinct structural patterns.

Table 2: Semantic matching statistics to reflect the "conflict" and "complementary" effects among SD3, SD, and DINOv2.

| Fusion | None (1984) | One (1974) | Two (2557) | All (5719) |
|---|---|---|---|---|
| Naive Con. | 6.4% | 40.9% | 88.3% | 99.5% |
| Ours | **12.7%** | **55.7%** | **90.9%** | **99.5%** |

Table 3: Comparisons under various conditions.

| | viewpoint diff. | | | scale diff. | | | truncation diff. | | | | occlusion diff. | | | |
|---|---|---|---|---|---|---|---|---|---|---|---|---|---|---|
| | easy | medi. | hard | easy | medi. | hard | none | source | target | both | none | source | target | both |
| SD3 | 61.2 | 46.7 | 43.9 | 56.5 | 54.0 | 47.7 | 57.1 | 51.4 | 50.4 | 48.8 | 56.7 | 49.5 | 49.2 | 50.0 |
| SD2 | 61.9 | 42.9 | 36.9 | 55.8 | 52.1 | 43.8 | 56.1 | 48.8 | 48.4 | 45.5 | 54.9 | 48.2 | 48.0 | 49.4 |
| DINOv2 | 55.7 | **58.1** | **57.4** | 58.5 | 57.0 | 50.6 | 59.4 | 54.4 | 52.9 | 50.1 | 57.2 | 55.5 | 55.3 | 58.0 |
| **Fuse2Match** | **70.7** | 56.8 | 50.7 | **65.5** | **63.5** | 58.6 | **65.7** | **62.1** | **61.2** | **58.6** | **65.1** | **61.0** | **61.0** | **61.7** |

*Model behavior in various cases.* Apart from evaluating the conflict and complementary effects among different foundation models, we provide additional experimental results on how these models behave in various cases. Specifically, we follow SPair-71k, which splits the image pairs in the test set into diverse variations, resulting in different levels (i.e., easy, medium, and hard) of viewpoint and scale, and various situations regarding truncation and occlusion. As shown in Table 3 (PCK@0.1),

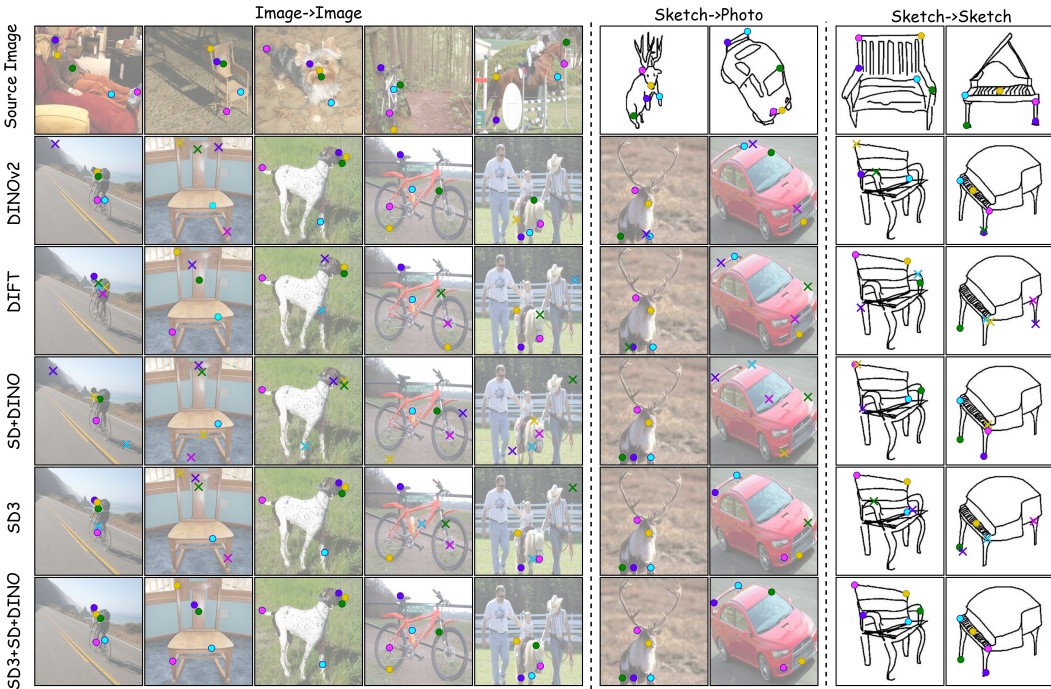

Figure 3: Comparison of semantic correspondence results using features from different pretrained models on the SPair-71k and PSC6K dataset. Key points in source images are color-coded. The incorrect matches are highlighted in cross mark in the corresponding color. Zoom in for the best view.

we can find out that (i) features from DiT-based SD3 alone do not exhibit a clear advantage over UNet-based SD, but SD3 indeed performs better on difficult cases on viewpoint and scale. (ii) DINOv2 outperforms both SD and SD3 generally, especially for hard cases, suggesting that DINOv2 offers accurate matching, while SD and SD3 excel at establishing rough correspondences. (iii) Utilizing our proposed feature fusion approach can significantly improve the performance of relying on any single model, verifying the efficacy of the confidence weight used in the process.

**Quantitative results on PF-Pascal and PSC6K.** As shown in Table 4, similar conclusions can be drawn that our integrated features of SD3+SD+DINO achieve the best performance across various matching thresholds on both the PF-Pascal and PSC6K datasets. Surprisingly, the performance of utilizing SD3 features alone can even surpass that of the combined SD+DINO features on the PF-Pascal dataset, highlighting its strong

Table 4: PF-PASCAL and PSC6K results.

| Method | PF-PSCAL | | | PSC6K | | |
|---|---|---|---|---|---|---|
| | 0.05 | 0.1 | 0.15 | 0.05 | 0.1 | 0.15 |
| DINOv1 | 55.6 | 74.2 | 81.6 | 38.98 | 65.05 | 78.76 |
| DINOv2 | 63.4 | 82.6 | 89.9 | 35.91 | 61.02 | 74.45 |
| DIFT | 69.0 | 82.2 | 88.1 | 35.80 | 58.67 | 72.37 |
| SD3 | 72.0 | 86.5 | 92.2 | 41.86 | 67.50 | 81.02 |
| SD+DINO | 68.1 | 85.7 | 91.5 | - | - | - |
| **Fuse2Match** | **78.6** | **90.6** | **94.6** | **49.62** | **74.31** | **86.28** |

capability for matching similar semantic locations between images. On the PSC6K dataset, we observe that SD3 features still outperform DIFT when matching sketch-photo pairs. Interestingly, while SD3 alone does not surpass the contrastively trained DINOv1/v2 models, the combination of features significantly improves the matching accuracy. This implies their complementary properties, which generalize effectively to image pairs with significant domain shifts using the integrated features.

**Qualitative Results.** To better understand the performance of different pretrained models in different matching scenarios, i.e., photo-to-photo, sketch-to-photo, and sketch-to-sketch, we present some qualitative comparison results in Figure 3. We can find that our proposed fused feature is capable of

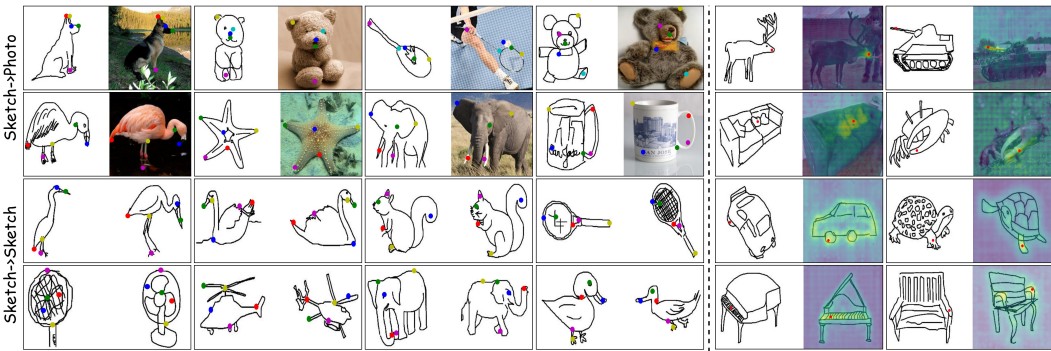

Figure 4: Visualization of semantic correspondence and matching heat maps by our fused SD3+SD+DINO feature on the PSDC6K dataset.

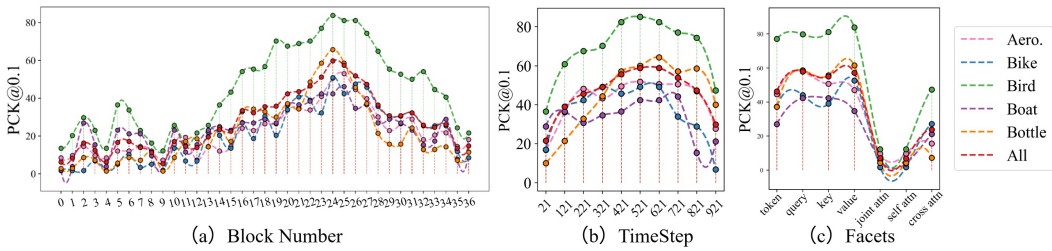

| (a) Block Number | (b) TimeStep | (c) Facets |

Figure 5: Ablation study results (PCK@0.1) on (a) block number, (b) timestep, and (c) facets of MM-DiT blocks evaluated on the first five categories of SPair-71k dataset.

more accurately identifying pixel pairs with similar semantics across different objects and varying viewpoints. Our approach demonstrates strong robustness to geometric transformations, such as aligning the left and right legs of a chair in the second row or the front and back wheels of a bicycle in the last row. In addition, our fused SD3+SD+DINO feature archives highly precise matching even between sketch-photo pairs or two sketches, demonstrating the robustness and generalizability of the feature. Moreover, we visualize more sketch2photo and sketch2sketch matching results coupled with attention heat maps on the PSC6K dataset. As shown in Figure 4, reasonable attention heat maps can be obtained, indicating the domain-agnostic semantic knowledge contained in the fused feature.

### 5.3 Ablation Study

We ablate several key hyperparameter choices of the feature extraction network (MM-DiT) on a subset of the SPair-71k training split, which is constructed by randomly selecting 10 photo pairs across all the 18 categories.

**Timestep and block number.** Similar to DIFT, the selection of timestep $t$ and block number is crucial for feature extraction. DIFT shows that a large timestep and earlier UNet upsampling layer yield more semantically-awere features. On the contrary, for extracting feature from SD3, our ablation results in Figure 5(a) and (b) reveal that a moderate timestep $t$ around 521 and later MM-DiT blocks around 24th yield better performances. In practice, we select three different timesteps, i.e., $t = [521, 621, 721]$, and conduct feature extraction from two attention blocks, namely 24 and 25, which have been proven to be effective in our case.

**Facets of MM-DiT.** We apply different facets in MM-DiT blocks serving as pixel-level features for semantic matching to study the impacts, involving the image tokens, queries, values, keys, self-

Table 5: Performance gains after performing confidence-aware feature fusion to various model combinations on a subset of SPair-71k dataset.

| Models | Aero. | Bike | Bird | Boat | Bottle | All |
|---|---|---|---|---|---|---|
| SD1.5+DINOv2 | +0.99 | +1.73 | -0.60 | -0.99 | +1.73 | +0.25 |
| SD3+SD2 | +0.25 | -0.75 | +0.87 | +0.27 | +0.48 | +0.27 |
| SD2+DINOv2 | +0.26 | -0.13 | +0.35 | +0.45 | +0.12 | +0.20 |
| SD3+DINOv2 | +0.84 | +1.50 | -0.02 | +2.28 | -0.07 | +0.74 |

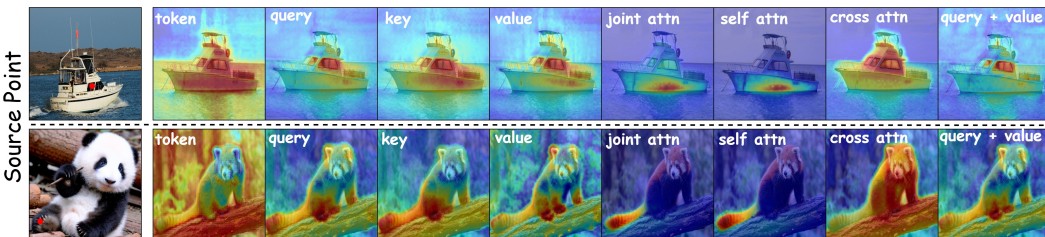

Figure 6: Visualization of matching probability maps using different MM-DiT facets. Features from `query` and `value` concentrate more precisely to the desired object parts.

attention, cross-attention, and joint attention. Experimental results in Figure 5(c) show that employing either the query or value as features for matching outperforms other options. In practice, we combine their strengths by concatenating query and value as the feature, which produces more accurate and sharper activations, as validated in Figure 6.

**Generalization of confidence-aware feature fusion.**   We evaluate the generalization capability of our proposed confidence-aware feature fusion strategy by applying it to different combinations of foundation models, involving SD3, DIFT (SD2/SD1.5), and DINOv2. As shown in Table 5, our proposed confidence-aware feature fusion achieves superior matching performance compared to naive concatenation, demonstrating its strong generalization ability across different model combinations.

# 6   Conclusions

We demonstrated that SD3, a transformer-based diffusion model, provides strong zero-shot semantic correspondence performance without task-specific fine-tuning. Through systematic exploration of SD3's architecture, we identified feature configurations that yield robust pixel-level matching. Moreover, we showed that SD3's global semantic features can be effectively complemented by SD and DINO's fine-grained representations. By introducing a confidence-aware fusion strategy, we adaptively integrate features from diverse models, significantly improving generalization across tasks such as photo-to-photo and sketch-to-photo matching.

**Limitations.**   The text prompt adopted in our approach is coarse-grained, which might be suboptimal and could limit the SD3 features for dense semantic matching, which demands fine-grained image understanding. Therefore, exploring ways to enhance SD3 features by prompt tuning further will be an important endeavor in the future.

# Acknowledgment

This work was supported by the Hainan Provincial Joint Project of Li'an International Education Innovation Pilot Zone (Grant No.624LALH008), BUPT Kunpeng&Ascend Center of Cultivation, NSFC (Grant No.61601042), the Program for Youth Innovative Research Team of BUPT (Grant No. 2023QNTD02), and the Super Computing Platform of BUPT.

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

# Appendix & Supplementary Materials

## A   Semantic Matching

In this section, we further present more visual results of semantic matching across different models in Figure 7 and 8. We also provide heatmap visualizations in Figure 9, which highlight the matching quality and indicate the domain-agnostic semantic knowledge contained in the fused feature.

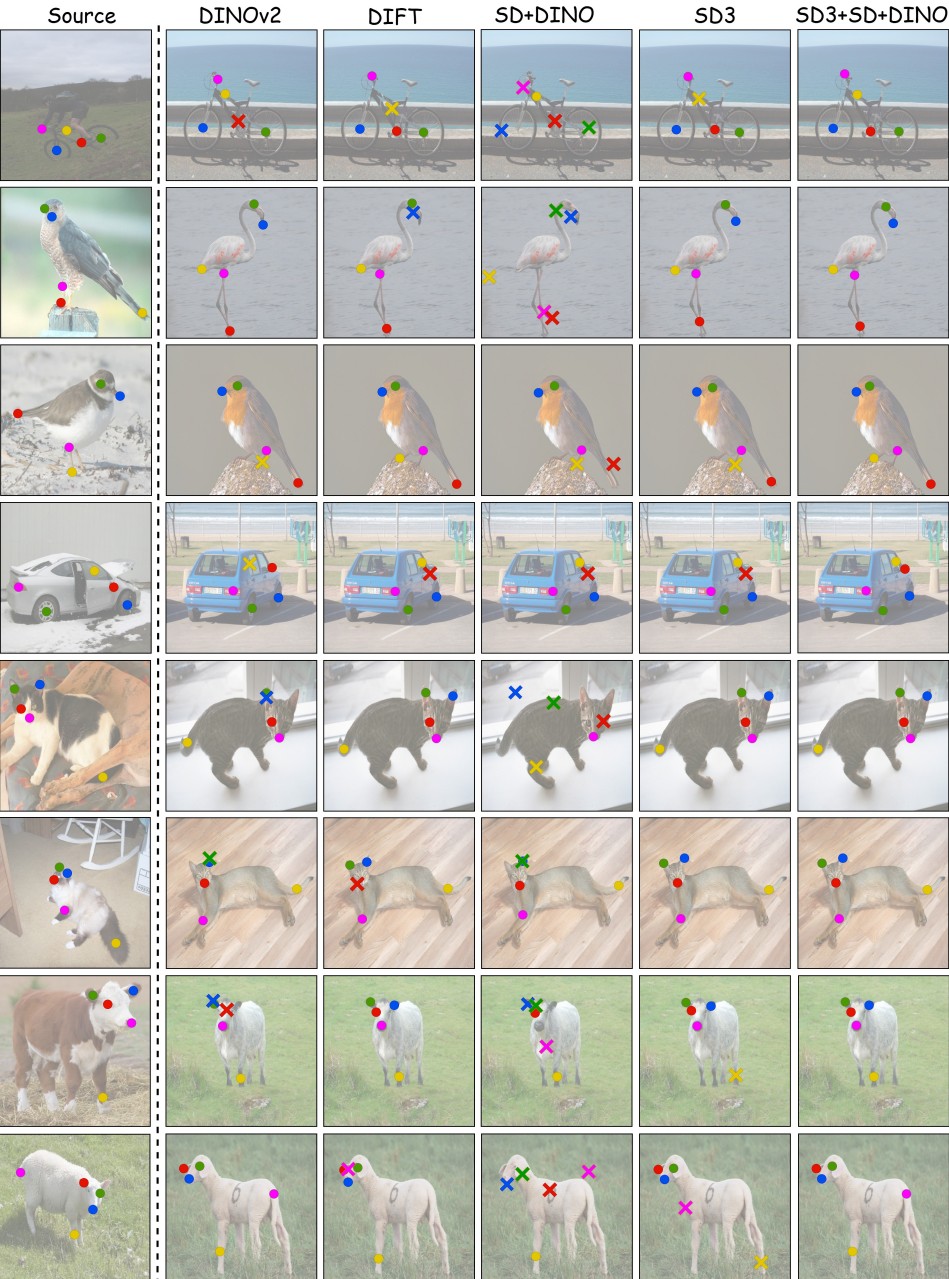

Figure 7: More comparison results on SPair-71k datasets.

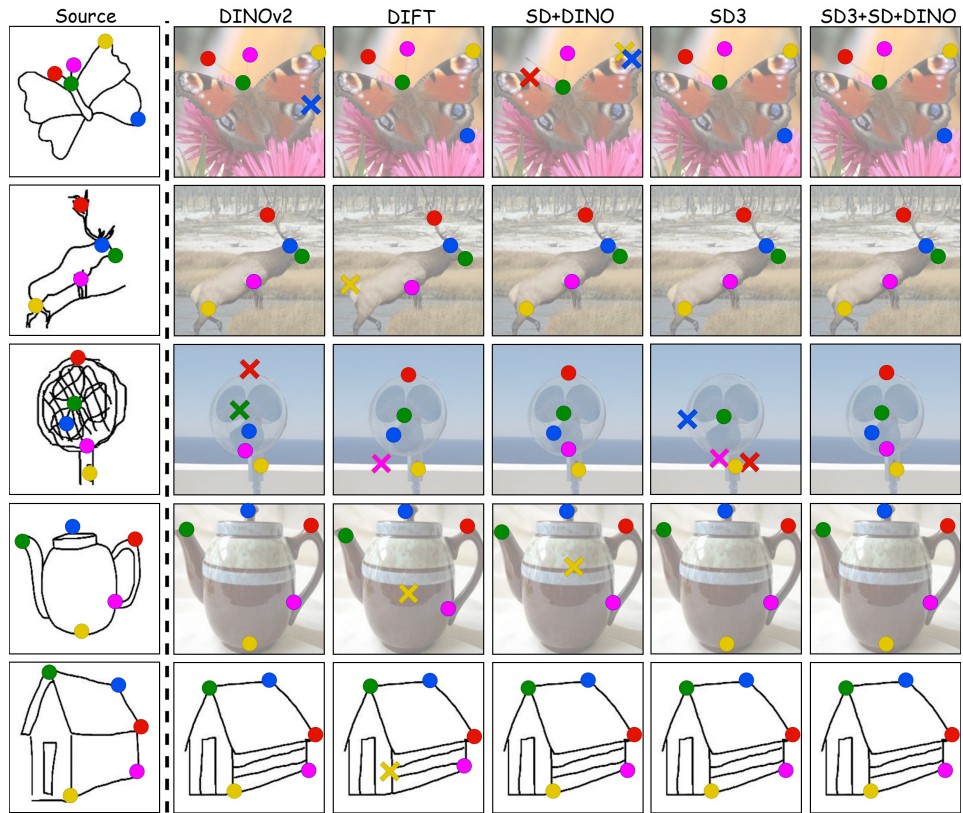

Figure 8: Comparison results for semantic correspondence between sketches and photos.

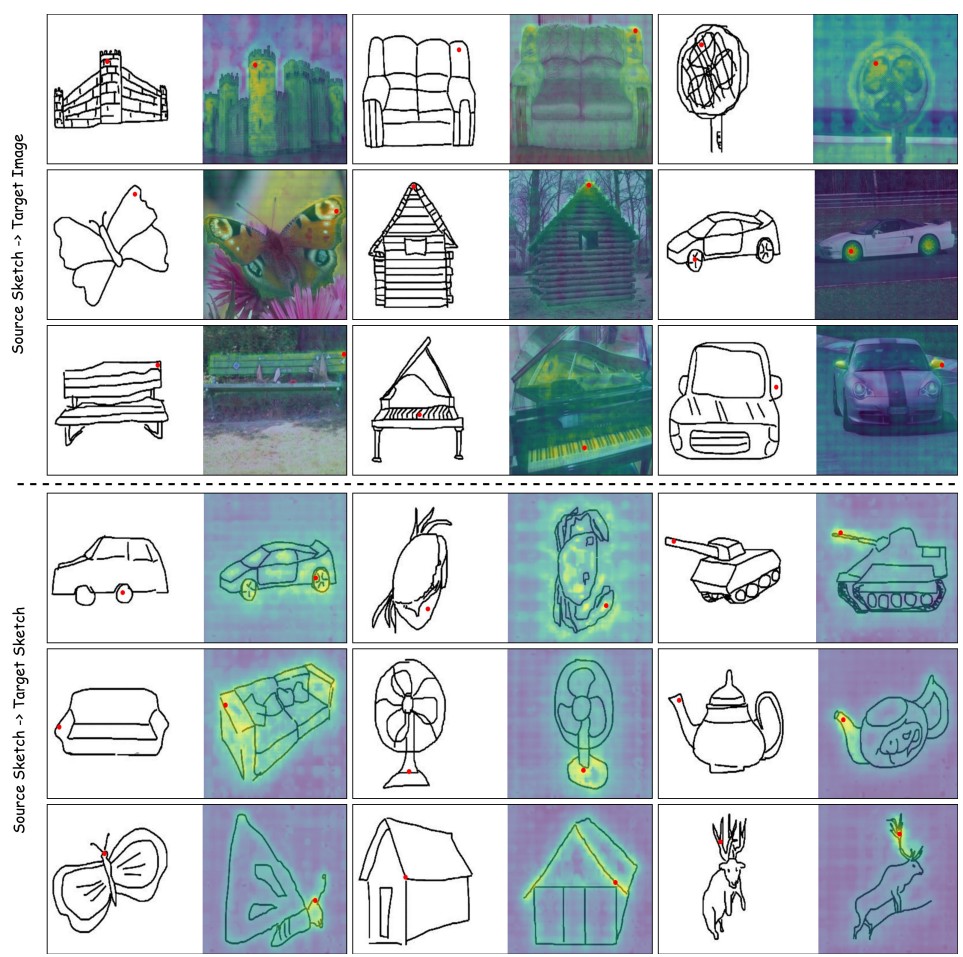

Figure 9: Heat maps of matching similarity between sketch-photo and sketch-sketch pairs.

# B   Instance-level Image Swapping

In this section, we further investigate the application of semantic matching to instance-level image swapping. Given the dense correspondence between two images established by our pixel matcher, we can transplant an instance from a source image onto the instance in a target image. The obtained new image preserves the identity of the source instance, while adapting to the pose or layout present in the target instance. Specifically, we replace the pixels of target image with the pixels in source image by nearest neighbor lookup, producing a swapped image. The swapped image is then refined using Stable Diffusion. As shown in Figure 10, the object in the swapped image resembles the pose and size of the object in target image (works well for sketch too) while the appearance comes from the source. After refinement using SD, high-quality images are obtained. We show more swap results in Figure 11and 12

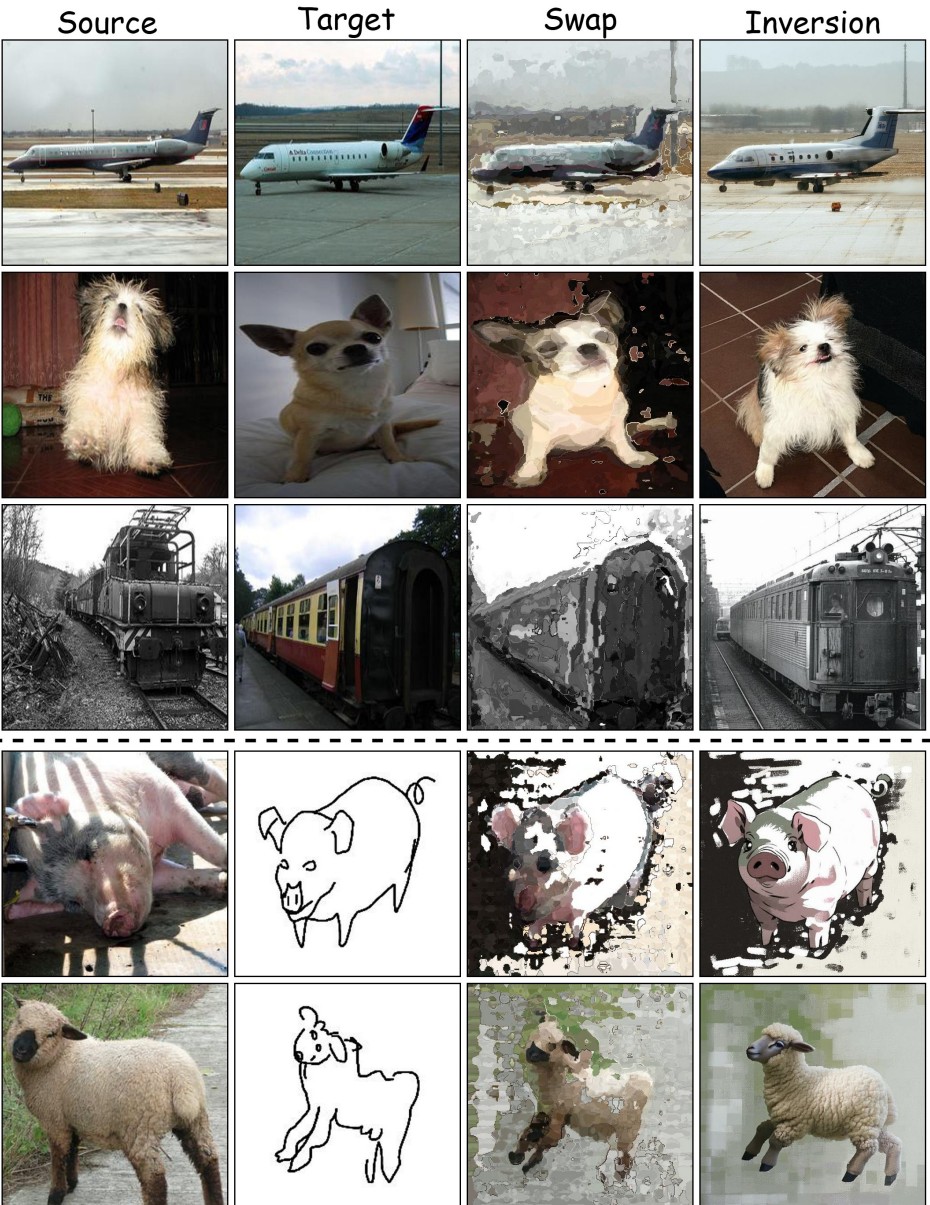

Figure 10: Instance swapping with Fuse2Matchfeatures. Swap: image after pixel replacement. Inversion: image refined by SD.

Source    Target    Swap    Inversion

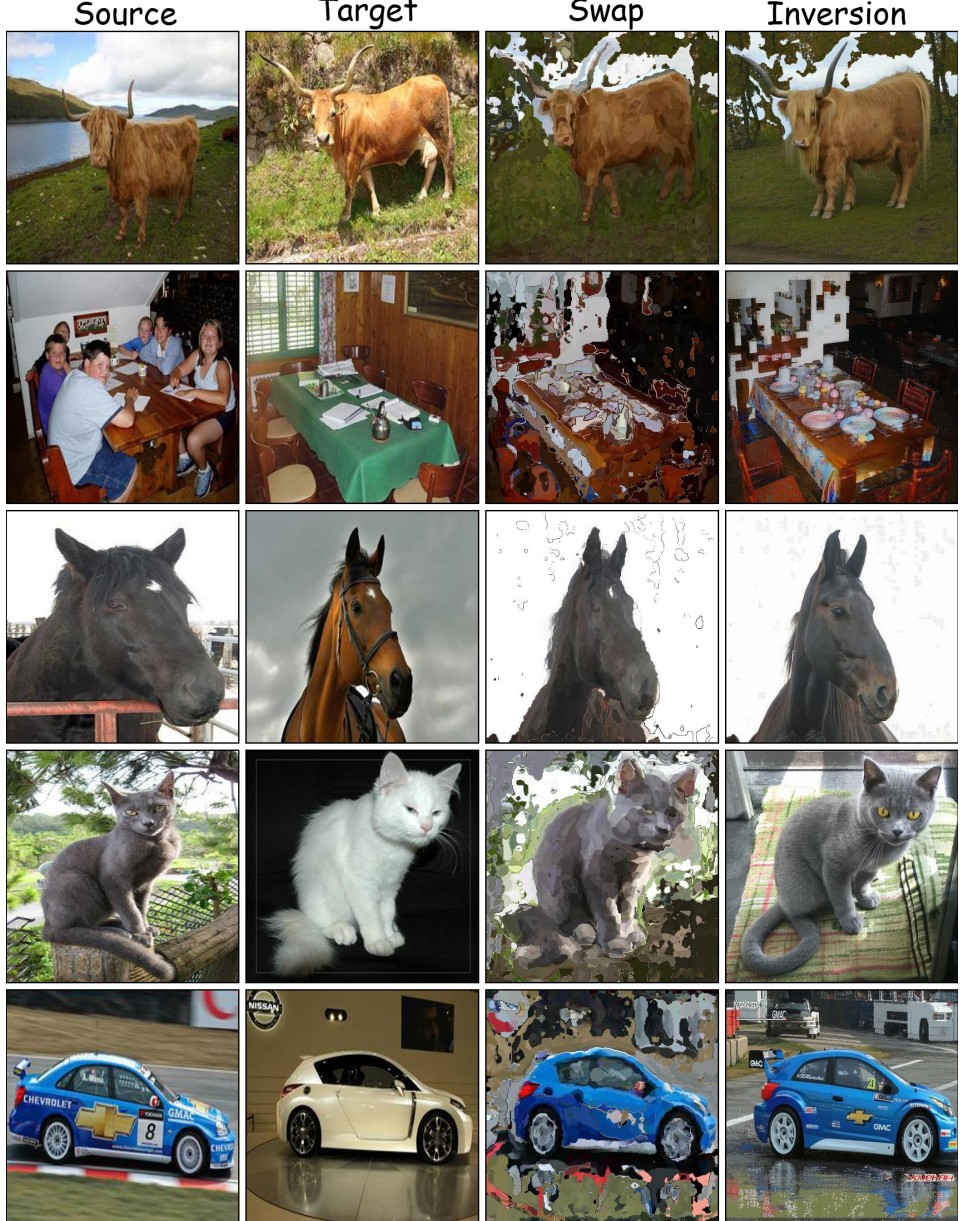

Figure 11: More instance swapping results between photos.

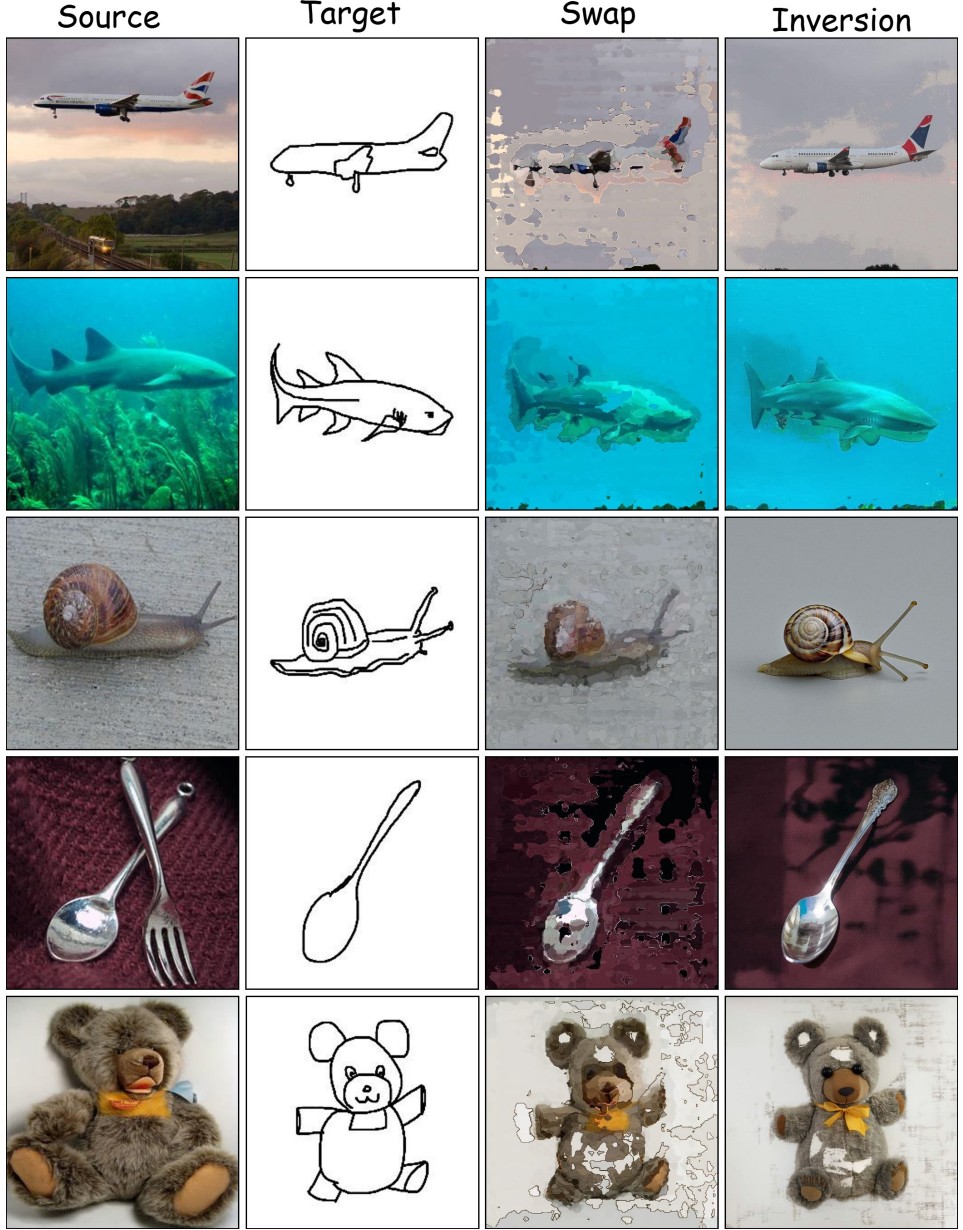

Figure 12: More instance swapping results from sketch to photo.

