# OpenReview forum: "Fuse2Match: Training-Free Fusion of Flow, Diffusion, and Contrastive Models for Zero-Shot Semantic Matching"
_NeurIPS.cc/2025/Conference — NeurIPS 2025 poster_

### Official Review · Reviewer_k3NR · 2025-06-26

**Clarity:** 3
**Significance:** 2
**Originality:** 2
**Rating:** 4
**Confidence:** 4

**Summary:**

This paper explores Stable Diffusion 3 (SD3) for the task of semantic correspondence. It makes two major contributions: First, it investigates the optimal configurations of the timestep in the diffusion process, the layer of the MM-DiT blocks, and the facets of the attention process to achieve the best zero-shot semantic matching. Second, it proposes a weighted combination method to combine features from SD3, SD, and DINOv2 and construct a hyperfeature to further improve the accuracy of zero-shot semantic matching. The paper compares the proposed method against the previous zero-shot methods and achieves state-of-the-art performance in this category.

**Questions:**

I believe this paper contains sufficient details, and I do not require anything substantial from the author. Maybe including the result of supervised methods using SD as the foundation in Table 1 helps improve the fairness of the Table.

**Ethical Concerns:**

["NO or VERY MINOR ethics concerns only"]

**Final Justification:**

In my opinion, this paper makes a solid contribution to the semantic correspondence field by investigating SD3 feature in this task. The author's rebuttal addressed the most of my concerns. Therefore, I would like to maintain my original rating (borderline accept)

**Limitations:**

Yes

**Paper Formatting Concerns:**

No formatting concern

**Quality:**

3

**Strengths And Weaknesses:**

**Strength**
1. This paper investigates SD3 for the semantic correspondence task, which is a novel attempt in this field.
2. Most of the claims are well supported by experimental evidence. The effectiveness of combining features from SD3, SD, and DINOv2 is demonstrated in Table 1, while the benefit of the proposed Confidence-aware Feature Fusion module is shown in the final two rows of the same table. Moreover, the feature fusion’s generalizability is strengthened by further experiments in Table 5, where it is applied to various model combinations.
3. Thorough details are given in the experiments, including the image size used for each method and hardware requirements for the method, which are useful to show the whole picture of the method.
4. The paper is well-written and easy to follow.

**Weakness**
1. While the use of SD3 is novel, the idea of combining features from SD and DINO is not. This concept was first introduced in SD+DINO [32], which this paper builds upon. As such, the core conceptual novelty is somewhat limited. That said, extending an existing research path in a meaningful way remains a valid and valuable contribution. I do not consider this overlap a significant weakness.
2. The paper selects two supervised methods, CATs++ and DHF, for comparison against these lines of work. However, using these works to create an impression that the proposed method can achieve a higher number in benchmarks over the supervised method is not fair. First, CATs++ employs ResNet101 as the backbone. ResNet101 is a much weaker general feature extractor than SD, so using CATs++ to highlight the performance of the proposed method is not fair. Second, although DHF employs SD as the backbone, the paper selectively ignores other supervised SD-based methods with much stronger performance, such as SD4Match [1] and GeoAware-SC [2]. Despite this unfairness, the proposed method is a zero-shot semantic matching method, and it should not be compared with supervised methods in the first place, as the supervised methods will most likely outperform zero-shot methods. Therefore, this flaw does not negatively impact the contribution of the paper.

**Conclusion**
This paper offers a meaningful contribution to the semantic correspondence literature by leveraging SD3 and proposing a confidence-aware fusion strategy. Despite some concerns about conceptual novelty and comparative evaluation, the experimental support is solid, and the direction is promising. I lean toward a borderline accept recommendation.

[1] Li, Xinghui, Jingyi Lu, Kai Han, and Victor Adrian Prisacariu. "Sd4match: Learning to prompt stable diffusion model for semantic matching." In Proceedings of the IEEE/CVF Conference on Computer Vision and Pattern Recognition, pp. 27558-27568. 2024.

[2] Zhang, Junyi, Charles Herrmann, Junhwa Hur, Eric Chen, Varun Jampani, Deqing Sun, and Ming-Hsuan Yang. "Telling left from right: Identifying geometry-aware semantic correspondence." In Proceedings of the IEEE/CVF Conference on Computer Vision and Pattern Recognition, pp. 3076-3085. 2024.

---

> ### Author Rebuttal · Authors · 2025-07-31
>
> We sincerely thank reviewer **k3NR** for the positive and encouraging feedback! We are glad that the contributions of our work are well recognized and valued.
> ***
> **Q1: To compare with supervised methods (CATs++ and DHF) to create an impression that the proposed method can achieve a higher number in benchmarks over the supervised method is not fair.**
>
> A1: We thank the reviewer for this constructive comment and fully agree that zero-shot methods should not be directly compared with supervised counterparts. Our primary goal in including CATs++ and DHF was to contextualize our method's performance, not to claim superiority over supervised methods. To avoid misleading readers, we will remove the comparisons to CATs++ and DHF in Table 1.
>
> **Q2: Maybe including the result of supervised methods using SD as the foundation in Table 1 helps improve the fairness of the Table.**
>
> A2: Thank you for the constructive suggestion. We agree that including results from recent supervised methods using SD as the foundation can improve the completeness and fairness of Table 1. In response, we will include the results of SD4Match (shown in below), a representative supervised method built upon SD, for comparison.
>
> | Method                  | Aero | Bike | Bird | Boat | Bottle | Bus  | Car  | Cat  | Chair | Cow  | Dog  | Horse | Motor | Person | Plant | Sheep | Train | TV   | All  |
> |-------------------------|------|------|------|------|--------|------|------|------|--------|------|------|-------|-------|--------|-------|--------|--------|------|------|
> | SD4Match                | 75.3 | 67.4 | 85.7 | 64.7 | 62.9   | 86.6 | 76.5 | 82.6 | 64.8   | 86.7 | 73.0 | 78.9  | 70.9  | 78.3   | 66.8  | 64.8   | 91.5   | 86.6 | 75.5 |
>
> As expected, SD4Match outperforms our training-free zero-shot approach, which is reasonable given that it benefits from task-specific supervision and fine-tuning. Nonetheless, our method remains competitive in several settings despite requiring no additional training, which highlights its efficiency and flexibility.
>
> **Q3: While the use of SD3 is novel, the idea of combining features from SD and DINO is not. This concept was first introduced in SD+DINO [32], which this paper builds upon. As such, the core conceptual novelty is somewhat limited. That said, extending an existing research path in a meaningful way remains a valid and valuable contribution. I do not consider this overlap a significant weakness.**
>
> A3: Thank you for the comment and kind words! While inspired by SD+DINO and DIFT, we want to emphasize that, our work goes beyond prior methods by tackling a more challenging setting: using rectified flow models with transformer backbones (SD3) for zero-shot correspondence. Due to SD3’s distinct flow-matching training and transformer architecture, feature extraction is highly non-trivial—naive choices underperform older UNet-based SD. We found that naive fusion hurts performance due to inconsistent feature distributions among SD3 and other foundation models. To address this, we introduce a confidence-aware fusion strategy that adaptively re-weights features per-pixel based on uncertainty, differing from uniform fusion in SD+DINO.

---

> > ### Comment · Reviewer_k3NR · 2025-08-03
> >
> > Thank you for your response. I will maintain my original rating and recommend borderline acceptance of this paper.

---

> > > ### Author Response · Authors · 2025-08-06
> > >
> > > Dear reviewer **k3NR**,
> > >
> > > Thank you for the time you've invested in reviewing our work. Your encouragement is greatly appreciated!
> > >
> > > Best regards,
> > >
> > > Authors

---

### Official Review · Reviewer_1cVT · 2025-07-01

**Clarity:** 3
**Significance:** 2
**Originality:** 2
**Rating:** 4
**Confidence:** 4

**Summary:**

This paper proposes a confidence-aware feature fusion strategy that re-weights contributions of different models based on prediction confidence scores. The key advantage of this strategy lies in its training-free nature. Experimental results on three semantic matching datasets demonstrate the effectiveness of the proposed method.

**Questions:**

1.Can you provide an analysis explaining why the performance gain on SPair-71k is limited?
2.How are other methods of calculating the prediction certainty? For example, entropy-based methods or variance.
3.The study should be enhanced by incorporating comparative analyses with advanced fusion methodologies beyond naive concatenation.

**Ethical Concerns:**

["NO or VERY MINOR ethics concerns only"]

**Final Justification:**

Most of my concerns are addressed, so I raise my rating to borderline accept.

**Limitations:**

See weeknesses.

**Paper Formatting Concerns:**

No concerns.

**Quality:**

2

**Strengths And Weaknesses:**

Strengths：
1.	The training-free architecture offers enhanced practical utility.
2.	The methodology and experimental procedures are presented with notable clarity.
3.	The comprehensive experimental framework conclusively demonstrates the method's effectiveness through multi-faceted validation.
Weaknesses：
1.	The proposed fusion strategy essentially extends weight adaptation upon naive concatenation, failing to introduce structural or theoretical innovations, thus constituting a marginal contribution.
2.	The confidence computation in the proposed method, while computationally straightforward, and the empirical demonstration of the fusion strategy's effectiveness notwithstanding, exhibit a conspicuous absence of theoretical justification and experimental validation for the predictive certainty quantification approach.
3.	The experimental section only compares the proposed method against naive concatenation, revealing marginal performance gains. More critically, it omits comparative analysis with state-of-the-art model fusion techniques.
4.	While experiments are conducted on three datasets, the method exhibits limited performance improvement on SPair-71k compared to substantial gains observed on the other two benchmarks. Crucially, the paper fails to account for this significant performance discrepancy across datasets.

---

> ### Author Rebuttal · Authors · 2025-07-31
>
> We sincerely thank reviewer **1cVT** for the critical and constructive feedback! The following provides our detailed responses to each of the concerns raised. We hope our clarifications resolve the outstanding issues and help strengthen the paper.
> ***
> **Q1: The proposed fusion strategy essentially extends weight adaptation upon naive concatenation, failing to introduce structural or theoretical innovations, thus constituting a marginal contribution.**
>
> A1: Thank you for the feedback. We respectfully disagree that our method constitutes a marginal extension of naive concatenation. While our fusion strategy may appear structurally simple, it addresses a critical but underexplored issue: the substantial feature inconsistency across foundation models due to their heterogeneous architectures and training objectives. Naive concatenation leads to suboptimal performance, as it treats all models equally regardless of their local reliability.
>
> In contrast, our confidence-aware fusion adaptively re-weights each model’s contribution in a per-pixel fashion, guided by matching uncertainty. This design departs from fixed-weight methods (e.g., SD+DINO) and shows superior flexibility and robustness across diverse model combinations.
>
> **Q2: The confidence computation in the proposed method, while computationally straightforward, and the empirical demonstration of the fusion strategy's effectiveness notwithstanding, exhibit a conspicuous absence of theoretical justification and experimental validation for the predictive certainty quantification approach.**
>
> A2: We thank the reviewer for the insightful comments. While our confidence metric is structurally simple, it is designed as a training-free, semantics-aware proxy to reflect local matching reliability. To support its effectiveness, we conducted a correlation analysis between the confidence scores and keypoint matching errors (measured by the Euclidean distance between predicted and ground-truth target points) on 100 randomly sampled cases using SD3, SD2, and DINOv2.
>
> As shown in the table below, we observe consistent moderate negative correlations (Pearson *r* ≈ –0.35 to –0.41, *p*-value < 1e–70), indicating that features with lower confidence scores tend to yield larger localization errors. This aligns with our design intuition that unreliable regions should be down-weighted in the fusion process. Here, Pearson *r* quantifies the linear relationship between the confidence score and matching error across individual features, while the associated *p*-value reflects the statistical significance of the correlation. We will incorporate these analyses into the revised version.
>
> **Table. Confidence–Error Correlation Analysis**
> | Feature  | Pearson *r* | *p*-value      |
> |---------|-----------|--------------|
> | SD3     | -0.3975   | 9.83e-99     |
> | SD2     | -0.4124   | 6.76e-107    |
> | DINOv2  | -0.3537   | 3.82e-77     |
>
> **Q3: The experimental section only compares the proposed method against naive concatenation, revealing marginal performance gains.**
>
> A3: Thank you for this valuable comment. Our work is primarily focused on training-free fusion of heterogeneous frozen backbones in zero-shot settings. And, it is important to note that in zero-shot scenarios with frozen encoders, even small yet consistent improvements across diverse settings are challenging to achieve. Our method shows reliable gains with negligible computational overhead, highlighting its robustness and plug-and-play usability.
>
> **Q4: More critically, it omits comparative analysis with state-of-the-art model fusion techniques.**
>
> A4: We appreciate the reviewer’s suggestion. We acknowledge the value of benchmarking against more sophisticated fusion frameworks (e.g., adaptive feature modulation [a], attention-based fusion [b], cross-modal distillation [c]). However, many of these methods rely on task-specific training or supervision, which diverges from our target: a **training-free, plug-and-play** fusion mechanism that emphasizes generality and applicability across diverse tasks and datasets. Exploring how our confidence-aware framework could be integrated with learnable fusion modules in a lightweight manner is a promising direction, and we plan to investigate this in future work.
>
> [a] Revisiting Feature Fusion for Image Classification. CVPR, 2021.
>
> [b] A Simple Framework for Contrastive Learning of Visual Representations. ICML, 2020.
>
> [c] Exploring Category-Agnostic Vision-Language Pretraining for Open-World Visual Understanding. CVPR, 2023.
>
> **Q5: While experiments are conducted on three datasets, the method exhibits limited performance improvement on SPair-71k compared to substantial gains observed on the other two benchmarks. Crucially, the paper fails to account for this significant performance discrepancy across datasets.**
>
> A5: We appreciate the reviewer for pointing out this important observation. Indeed, the performance gain on SPair-71k is less prominent compared to the other two benchmarks. We attribute this discrepancy to the inherent challenges of SPair-71k, which involves larger viewpoint changes, stronger non-rigid deformation, and noisier annotations. These factors significantly degrade the performance of all pretrained models involved (e.g., SD3, DINOv2), thus limiting the headroom for additional gains through fusion. Nevertheless, our method still consistently outperforms individual models and naive fusion strategies on SPair-71k, as reported in Table 3 in our paper. This indicates that even under such adverse conditions, our confidence-aware fusion remains effective and robust. We will clarify this point more explicitly in the revised version.
>
> **Q6: How are other methods of calculating the prediction certainty? For example, entropy-based methods or variance.**
>
> A6: Thank you for the suggestion. We agree that uncertainty estimation methods such as entropy or variance have been widely used, especially in classification tasks. In our case, however, the proposed confidence score is computed based on the feature-level consistency across multiple pretrained models, which is **more suitable for training-free fusion and semantic matching tasks where no explicit distribution is available**.
>
> In contrast, entropy- or variance-based methods typically require either class probability distributions or Bayesian/posterior modeling, which is non-trivial in our zero-shot or training-free setting. Nevertheless, to evaluate their applicability, we conducted preliminary comparisons with entropy- and variance-based certainty scores adapted to our context. As shown in the table below, our method consistently outperforms these alternatives, highlighting its effectiveness in guiding fusion without requiring distributional assumptions.
>
> | Feature Fusion Strategy           | PCK\@0.1 |
> |--------------------|------------------------|
> | Entropy-based            | 60.5                   |
> | Variance-based           | 66.8                   |
> | Ours     | 68.8                  |

---

> > ### Comment · Area_Chair_Y4Wt · 2025-08-06
> >
> > Dear Reviewer,
> > Could you please read the rebuttal, update your final review, and share any remaining follow-up questions, if any? Also, kindly acknowledge once this is done.
> > Thank you.

---

> ### Author Response · Authors · 2025-08-09
>
> Dear reviewer **1cVT**,
>
> As the author–reviewer discussion phase is approaching its end, we wanted to follow up to see if our rebuttal has addressed your concerns. Your feedback would be greatly appreciated, as it will help us ensure that we have fully clarified any remaining issues.
>
> Thank you again for your time and constructive input throughout the review process.
>
> Best regards,
>
> Authors

---

### Official Review · Reviewer_Aycd · 2025-07-02

**Clarity:** 3
**Significance:** 3
**Originality:** 3
**Rating:** 4
**Confidence:** 4

**Summary:**

This work further combines the Stable Diffusion 3, Stable Diffusion and DINO v2 called E-SD^3 for semantic correspondence. In E-SD^3, a multi-level fusion scheme  is designed to extract discriminative features and a confidence-aware feature fusion strategy to handle the issue of nconsistent distributions across models. This methos outperforms state-of-the-arts on three SC datasets.

**Questions:**

See weaknesses.

**Ethical Concerns:**

["NO or VERY MINOR ethics concerns only"]

**Final Justification:**

Thanks for the detailed responses. My concerns are solved completely so I keep the rating and tend to accept this paper.

**Limitations:**

yes

**Paper Formatting Concerns:**

None.

**Quality:**

3

**Strengths And Weaknesses:**

Strengths

1. E-SD^3 performs well on all the datasets especially in the matching tasks of skeleton and image (PSC6K).

2. The designed feature fusion strategy is effective to make full use of strong backbones.

Weakness

1. In table 1, best and second best should be labeled for each settings instead of labeled together.

2. How do the three backbones contribute to the SC result after fusion? How is the effect of using the same fusion idea to process SD and DINO2?

3. Figure 4. Why are there multiple highlighted regions for geometrically similar objects such as wheels and chair arms in matching heat maps?

---

> ### Author Rebuttal · Authors · 2025-07-31
>
> We sincerely thank reviewer **Aycd** for the insightful feedback! The suggestions have helped us better refine the presentation and further clarify key aspects of the work.
> ***
> **Q1: How do the three backbones contribute to the SC result after fusion?**
>
> A1: Thank you for the insightful question. To understand the contributions of different backbones in the fused representation, we analyzed the confidence-aware fusion weights across the test set. As shown in the table below, we empirically observed that SD3 tends to contribute more in semantically complex scenes, likely due to its robust but coarser correspondence capability. DINOv2 appears to play a larger role in structure-sensitive regions (e.g., vehicles), possibly benefiting from its precise yet noisier features. Meanwhile, SD provides relatively more stable cues in low-texture regions. While these insights are based on limited observations and preliminary statistics, they are consistent with the feature behaviors illustrated in Figure 2 and offer useful insights into the complementary strengths of each backbone. We will add more qualitative and quantatitive results accordingly in our revised version.
>
> **Table: Fusion weights analysis.** Here, *Complex* refers to scenes with multiple objects or cluttered backgrounds (e.g., ''pottedplant'' and ''person''). *Structure-sensitive* targets are man-made objects with stable geometry (e.g., ''aeroplane and ''motorbike''). *Low-texture* denotes smooth-surfaced objects lacking rich local details (e.g., ''tv-monitors'' and ''bottle'').
>
> |                      | SD3    | SD2    | DINOv2 |
> |----------------------|--------|--------|--------|
> | Complex              | **0.3511** | 0.3417 | 0.3070 |
> | Low-texture          | 0.3300 | **0.3433** | 0.3265 |
> | Structure Sensitive  | 0.3281 | 0.3242 | **0.3486** |
>
> **Q2: What is the effect of using the same fusion idea to process SD and DINO2?**
>
> A2: Thank you for the question. In fact, we have included the comparison results of fusing SD(1.5) with DINOv2 using different fusion methods (i.e., naive concatenation vs confidence-aware fusion) in Table 5, which suggests that our confidence-aware fusion strategy can be effectively generalized to other settings.
>
> **Q3: Why are there multiple highlighted regions for geometrically similar objects in Figure 4.**
>
> A3: Sorry for the confusion. In Figure 4, given a query point in the source image, we compute its feature similarity to all points in the target image, resulting in a dense attention map. This map is then color-coded by the similarity scores, where warmer colors indicate higher similarity. As such, multiple highlighted regions may appear when the model assigns moderately high attention to several geometrically similar areas (such as the wheels and chair arms you noticed). However, only the point with the highest attention score (e.g., the left chair aim which has both geometry and location correct) is selected for the final match. The purpose of the visualization is to show the entire attention distribution, rather than just the final matched point. We will further clarify this point in our revised version.
>
> **Q4: Format issue in Table 1**
>
> A4: Thank you! We will fix it accordingly.

---

> > ### Comment · Area_Chair_Y4Wt · 2025-08-06
> >
> > Dear Reviewer,
> > Could you please read the rebuttal, update your final review, and share any remaining follow-up questions, if any? Also, kindly acknowledge once this is done.
> > Thank you.

---

> > ### Comment · Reviewer_Aycd · 2025-08-07
> >
> > Thanks for the detailed responses. My concerns are solved completely so I keep the rating and tend to accept this paper.

---

### Official Review · Reviewer_S5hJ · 2025-07-03

**Clarity:** 2
**Significance:** 2
**Originality:** 2
**Rating:** 4
**Confidence:** 1

**Summary:**

The paper explores whether the recently released Stable Diffusion 3 (SD3) transformer backbone, trained with a rectified-flow objective, contains spatial cues useful for zero-shot semantic correspondence, a task previously dominated by UNet-based Stable Diffusion 2 and contrastive ViTs such as DINO. After an exhaustive greedy search that shows semantic signals are scattered across mid-noise timesteps and late MM-DiT layers, the authors build a multi-level SD3 descriptor and then fuse it with SD 2.1 and DINO v2 through a confidence-aware weighting that amplifies high-certainty pixels, defined as the gap between the maximum and mean of each similarity vector (L185-L190). This training-free ensemble, dubbed E-SD3, improves PCK@0.1 from 59.8 for SD3 alone to 68.8 on SPair-71k (L234-L244) and beats prior tuning-free baselines on PF-Pascal and PSC6K as well (L289-L299), while requiring only inference-time feature extraction.

**Questions:**

Could the authors clarify the wall-clock latency and peak memory when processing a pair, and whether those numbers include the VAE encoder-decoder passes demanded by SD3? Have they tried a lightweight student network or linear adapter trained to emulate the fused representation, and if so how does its accuracy-vs-speed trade-off look?
Why was the particular confidence metric chosen over entropy-based or temperature-scaled alternatives, and does its effectiveness persist if all three backbones are similarly uncertain? Given that the greedy search for timesteps, layers, and facets was conducted on SPair-71k training data (L318-L324), can the authors report performance on a held-out dataset unseen during that search to rule out subtle over-tuning?

**Ethical Concerns:**

["NO or VERY MINOR ethics concerns only"]

**Final Justification:**

All concerns were adaquately addressed in the rebuttal, raising final recommendation to borderline accept.

**Limitations:**

Please see questions and weaknesses. I am not a specialist in this sub-field, so my current recommendation is tentative and could shift after reading the authors’ rebuttal.

**Paper Formatting Concerns:**

I don't notice any major formatting issues

**Quality:**

3

**Strengths And Weaknesses:**

The work is timely, being the first to repurpose SD3 for dense matching (L63-L65), and it demonstrates that transformer-based diffusion features can complement both UNet-diffusion and contrastive ViT signals, yielding clear accuracy gains without any fine-tuning.

However, I suppose those gains arrive at a steep computational cost as a single image produces a SD3 map that must be compressed just to fit on an A100-40 GB GPU (L230-L232). Memory footprint and inference time comparison seems required. “confidence-aware fusion” sounds a hand-crafted re-weighting based on the max-minus-mean gap (L185-L190), a surrogate for saliency that closely mirrors the mutual-nearest-neighbor or peak-ratio heuristics. Apart from being the first to adaping SD3 for matching, what is the paper's main technical contribution?

---

> ### Author Rebuttal · Authors · 2025-07-31
>
> We sincerely thank reviewer **S5hJ** for the critical and constructive feedback! Below, we provide point-by-point responses to the specific questions and concerns raised, and we hope these clarifications adequately address the issues.
> ***
> **Q1: Performance gains arrive at a steep computational cost.**
>
> A1: We thank the reviewer for pointing out this important aspect. Raw SD3 features are large indeed, and we adopt a compression step not merely as a workaround but as a core design to promote alignment across heterogeneous models and reduce resource consumption. To quantify this following your suggestion, we report memory and latency comparisons in the Table bellow. With compression (we concatenate all per-pixel features and apply PCA to reduce the dimension to 1K), SD3 inference requires ~3.5 GB GPU peak memory (less than DINOv2 and SD2) and has manageable matching runtime (\~0.03s per pair, 9x faster than the SD3 only using matrix factorization) while still maintaining competitive results (PCK@0.1 at 58.98 on SPair-71k), making it feasible for real-world, training-free zero-shot scenarios. We agree that further improvements in efficiency (e.g., via distillation) are promising directions for future work.
>
> Table: Comparison of Inference Time (Including VAE encoder) and Peak Memory Usage per Pair across Different Models
> | Model             | Feature Extraction Time (s) | Dense Matching Time (s) | Peak Memory (GB) | PCK\@0.1|
> |------------------|-----------------------------|--------------------------|------------------|------------------|
> | SD3              | 0.530                       | 0.312                    | 29               | 59.48
> | SD3 (compressed) | 0.530                       | 0.039                    | 3.5              | 58.98
> | SD2              | 0.294                       | 0.040                    | 4.7              | -
> | DINOv2           | 0.022                       | 0.054                    | 3.8              | -
>
> **Q2: Confidence-aware fusion strategy is a surrogate for saliency that mirrors the mutual-nearest-neighbor or peak-ratio.**
>
> A2: Thank you for the inspiring observation. We conducted additional experiments using peak-ratio [a]—the ratio between the top-1 and top-2 values in the similarity vector M (Eq. 5)—as a confidence measure. While conceptually intuitive, peak-ratio yields lower performance (PCK\@0.1: 66.94) than our proposed max-minus-mean strategy (PCK\@0.1: 68.8). We attribute this gap to peak-ratio’s sensitivity to local noise, i.e., when multiple candidates have similar similarity scores, the ratio becomes unstable and less indicative of true confidence. In contrast, our max-minus-avg captures global contrast by penalizing non-peak similarities collectively, leading to more robust fusion.
>
> Regarding the resemblance to mutual nearest neighbor (MNN) [b], we appreciate the reviewer's perspective but currently do not see a direct conceptual link. MNN is primarily used for establishing reliable correspondences between two sets, while our strategy operates at the feature aggregation level. We would be grateful if the reviewer could elaborate on the intended analogy.
>
> [a] Ensemble Learning for Confidence Measures in Stereo Vision. CVPR, 2013.
>
> [b] Best-buddies similarity for robust template matching. CVPR, 2015.
>
> **Q3: Have you tried a lightweight student network or linear adapter trained for feature fusion?**
>
> A3: Thank you for the insightful suggestion. Our work specifically focuses on a training-free fusion strategy that directly integrates heterogeneous pre-trained features without any fine-tuning or additional modules. While training a lightweight student network or linear adapter is indeed a promising direction for future work, it falls beyond the scope of our current objective.
>
> **Q4: Why was the particular confidence metric chosen over entropy-based or temperature-scaled alternatives, and does its effectiveness persist if all three backbones are similarly uncertain?**
>
> A4: We appreciate the reviewer’s insightful question. Our confidence metric was selected due to its training-free nature and effectiveness in reflecting semantic saliency across heterogeneous feature spaces. Unlike entropy-based or temperature-scaled metrics which rely on probability distributions or require tuning, our method operates directly on normalized feature activations and scales efficiently across diverse models. We have also experimented with entropy-based confidence metric. As shown in the table below, our confidence metric yields better results.
>
> When all backbones are similarly uncertain, the confidence scores tend to flatten, leading to more balanced fusion weights according to eq.6. We believe this acts as a natural safeguard against overfitting to unreliable representations.
>
> Table: PCK\@0.1 results on SPair-71k with different feature fusion strategies.
>
> | Strategy           | PCK\@0.1 |
> |--------------------|------------------------|
> | Entropy-based            | 60.5                   |
> | Ours     | 68.8                  |
>
> **Q5: Performance on held-out datasets**
>
> A5: Sorry for the confusion. In fact, our search for fusion hyperparameters (i.e., timestep, layer, and facet) was conducted exclusively only on the SPair-71k training split, with the test split held out for final evaluation (as reported in Table 1 of the paper). Moreover, to assess the generalizability of the selected configuration, we directly applied the same hyperparameters to PF-PASCAL and PSC6K without any re-tuning. The consistent performance gains observed in Table 1 and Table 4 in the paper suggest that our method is not overfitted to the SPair-71k training set and that the selected configuration is robust across datasets.

---

> > ### Comment · Reviewer_S5hJ · 2025-08-06
> > **Response**
> >
> > Thank you for the authors' effort in the rebuttal. All concerns are addressed. I'm raising my final recommendation to borderline accept.

---

> > > ### Author Response · Authors · 2025-08-06
> > >
> > > Dear reviewer **S5hJ**,
> > >
> > > We are glad that your concerns have been addressed! Thank you for taking the time to review our rebuttal and for the encouraging recommendation.
> > >
> > > Best regards,
> > >
> > > Authors

---

### Decision · Program_Chairs · 2025-09-17

**Decision:**

Accept (poster)

**Comment:**

This paper addresses the task of semantic correspondences and investigates the quality of flow-matching features (in the form of SD3) for the task. The paper finds that the features that are useful for the task are spread across timesteps and designs a fusion technique to gather these features. The paper then creates an ensemble of feature sets and demonstrates strong performance on semantic correspondence benchmarks.

The reviews were positive for this paper (4 borderline accept). The reviewers commended the performance of the approach as well as the clean analysis and presentation. The main drawback was seen as the relatively weak technical novelty of the paper. I generally agree with all these points but also agree with the authors that there is value in understanding the difference in features of different styles of generative models.

I advocate for acceptance.